

# Heat waves in Africa 1981-2015, observations and reanalysis

G. Ceccherini[1], S. Russo[2], I. Ameztoy[1], A. F. Marchese[3] and C. Carmona-Moreno[1]

[1]European Commission, Joint Research Centre (JRC), Institute for Environment and Sustainability (IES), Water Unit, Via E. Fermi 2749, 21027 Ispra, Italy

5 [2]European Commission, Joint Research Centre (JRC), Institute for the Protection and Security of the Citizen (IPSC), Financial and Economic Analysis Unit, Via E. Fermi 2749, 21027 Ispra, Italy}

[3]Università degli Studi di Catania Dipartimento di Fisica e Astronomia, Via Santa Sofia 64 - 95123 Catania

*Correspondence to*: C. Carmona-Moreno (cesar.carmona-moreno@jrc.ec.europa.eu), tel. (+39) 0332 78 9654, fax (+39) 0332 78 9073

10 **Abstract.** The purpose of this article is to show the extreme temperature regime of heat waves across Africa over recent years (1981–2015). Heat waves have been quantified using the Heat Wave Magnitude Index daily (HWMId), which merges the duration and the intensity of extreme temperature events into a single numerical index. The HWMId enables a comparison between heat waves with different timing and location, and it has been applied to maximum and minimum temperature records. The time series used in this study have been derived from: 1) observations from the Global Summary of the Day (GSOD); and 15 2) *reanalysis* data from ERA-INTERIM. The analysis show an increasing numbers of heat waves of both maxima and minima temperatures in the last decades. Results from heat wave analysis of maximum temperature (HWMId$_{tx}$) indicate an increase in intensity and frequency of extreme events. Specifically, from 1996 onwards it is possible to observe HWMId$_{tx}$ spread with the maximum presence during 2006-2015. Between 2006 and 2015 the frequency (*spatial coverage*) of extreme heat waves had increased to 24.5 observations (*60.1% of land cover*) per year, as compared to 12.3 (*37.3% of land area*) per year in the period 20 from 1981 to 2005 for GSOD stations (*reanalysis*).

## 1 Introduction

Africa is considered one of the most vulnerable regions to weather and climate variability (Solomon et al., 2007). Under this perspective, variability and changes in extreme temperature regimes present a considerable challenge: different aspects of the 25 regime of extreme temperatures - both spatially and temporally - are still lacking. Such information is one of the central issues within the global change debate, since it is necessary to assess the impacts of climate change on human and natural systems and to develop suitable adaptation and mitigation strategies at country level.

In order to analyse extreme temperature regimes, daily records are needed. To this end, the Global Surface Summary of the Day (GSOD) meteorological dataset has been employed. GSOD is a compilation of daily meteorological data produced by the 30 National Climatic Data Center, available from 1929 to present, which displays a reasonably dense spatial coverage across Africa. However, a general caveat with the GSOD dataset is the limit imposed by its sparse gauge network. There are many





regions, especially across Central Africa where the absence of temperature records precludes a comprehensive and robust analysis. To circumvent this limitation, also *reanalysis* data has been used. *Reanalysis* is a combination of observations and climatological models through data assimilation systems to produce a single, uniform global data set (Kalnay et al., 1996), thus enabling a homogeneous coverage of Africa.

The magnitude of heat waves for both observations and *reanalysis* is quantified on annual basis by means of the Heat Wave Magnitude Index daily (HWMId, Russo et al., 2015) for the period 1981-2015 across Africa. The HWMId has been applied to maximum and minimum temperature.

The objective of this paper is to analyse African heat wave regime and identify the most important of heat waves during 1981-2015. These analyses draw attention to the spatial distribution of temperature extremes and their temporal evolution in the past

decades, still largely unknown.

Some early exploratory research using similar methodology for South America showed some promising preliminary results (Ceccherini et al., 2015). In this paper, the heat wave classification scheme has been consolidated and improved (Russo et al., 2015) and both observations and *reanalysis* datasets have been employed.

## 2 Materials and Methods

### 2.1 Materials

The time series of temperature used in this study have been derived from: 1) observations; and 2) *reanalysis*. Global Surface Summary of the Day (GSOD) is the dataset of observations. GSOD records, produced by the National Climatic Data Center, are mainly recorded at international airports and include maximum and minimum values of temperature. GSOD has been already employed to assess heat waves magnitude at global (Mishra et al., 2015) and local (Ceccherini et al., 2015) scale.

Historical data are generally available for the last 80 years, with data from 1973 onwards being the most complete. The total number of GSOD stations available across Africa is equal to 958. However, only 260 of them satisfy the conditions needed to calculate heat wave magnitude indices, as further described in the Method section.

ERA-INTERIM (Dee et al., 2011) is the dataset of *reanalysis*. *Reanalysis* has been increasingly used to address a variety of climate-change issues and has by now become an important method in climate change research (Fan and van den Dool, 2004;

Marshall and Harangozo, 2000; Uppala et al., 2005). ERA-INTERIM is a *reanalysis* product of the European Centre for Medium-Range Weather Forecasts (ECMWF) available from 1979 and continuously updated in real time. The data assimilation system used to produce ERA-INTERIM is based on a 4-dimensional variational scheme (4D-Var) with a 12-hour analysis window (for further information on 4D-VAR see Courtier et al., 1994). The spatial resolution of the data set is 0.75° (i.e., approximately 80 km at the equator). ERA-INTERIM allows a consistent spatial and temporal resolution over 3 decades,

incorporating millions of observations into a stable data assimilation system that would be nearly impossible for an individual to collect and analyse separately. On the other hand, observational constraints, and therefore *reanalysis* reliability, can





considerably vary depending on the location, time period, and variable considered, thus introducing spurious variability and trends.

For both observations and *reanalysis* dataset the timespan considered in this study refers to the period January 1, 1981 - June 30, 2015.

## 2.2 Methods

In this paper the Heat Wave Magnitude Index daily (HWMId), recently defined by Russo et al. (2015), has been employed to detect African heat waves for the period 1981-2015. The HWMId is a simple numerical indicator that takes both the duration and the intensity of the heat wave into account. Basically, the magnitude index sums excess temperatures beyond a certain normalized threshold and merges durations and temperature anomalies of intense heat wave events into a single indicator. The HWMId is an improvement on the previous Heat Wave Magnitude Index (i.e., HWMI, Russo et al. 2014) and it is able to overcome HWMI limitations in assigning magnitude to very high temperatures in a changing climate (for further detail see (Russo et al., 2014, 2015)).

The HWMId computations requires at least a 30-year time series of daily temperature records. GSOD stations with less than 30 years records and with more than 30% of gaps have been excluded from our analysis (for further details see Ceccherini et al. 2015). As a result, 260 GSOD stations out of 958 satisfy these conditions. Figure 1 shows the spatial distribution of the 260 temperature stations satisfying these conditions.

Heat waves are computed using: 1) maximum (hereafter $HWMId_{tx}$), and 2) minimum (hereafter $HWMId_{tn}$) daily temperature, giving thus complementary information respectively on warm day and night conditions. HWMId has been computed on annual basis: 1) for each GSOD station, and 2) for the entire spatial domain of the gridded ERA-INTERIM dataset across Africa. HWMId scale is defined following the classification scheme proposed by Russo et al. (2015), where the magnitude scale is assigned with values in $[0, +\infty[$ and not in a bounded interval as for the HWMI. Specifically, HWMId scale depends on the inter quartile range of 30-year annual maximum temperature within the reference period 1981–2010 (for further details see Russo et al. 2015).

Heat waves generally occur between December and January in the southern hemisphere and between June and July in the northern. In order to avoid splitting event occurrences that happen within a regular calendar year, starting and ending dates have been redefined accordingly. Therefore, the HWMId computation starts on January 1, 1981 and ends on December 31, 2014 in the Northern hemisphere. Similarly, the HWMId computation starts on July 1, 1981 and ends on June 30, 2015 in the Southern hemisphere (for further information see HWMId function in Gilleland and Katz 2011).

Note that the Southern hemisphere 6-month time shift might cause temporal inconsistency in the dataset: the time span is 34 years in the Southern hemisphere and 35 in the Northern one. However, starting our analysis from 1980 would have further reduced the number of available GSOD stations from 260 to ~220, thus exaggerating the already patchy spatial distribution of the observational network.





## 3 Results

Figure 2 and 3 display the maximum value in 5-year periods of the HWMId of GSOD observations from 1981 to 2015 for maximum and minimum temperature, respectively.

There is a clear indication that both intensity and spatial distribution of heat waves of maximum temperature are increasing.

Specifically, from 1996 onwards it is possible to observe a positive trend in heat waves' magnitude and spread across Africa, with the maximum presence during 2011-2015. HWMId$_{tx}$'s frequency is below 40 events per 5-year period until 1995, and then increases. This is noticeable from the analysis of the histograms at the bottom of the figure: the maximum value of the occurrence of heat waves per 5-year period rises from ~40 to ~120 events from 1981-1985 to 2011-2015. Such observation fits well with what was found in Fontaine et al. (2013) across Northern Africa, i.e. heat wave frequency is increasing after

1997, with a mean frequency multiplied by 2 or 3.

Despite the generally high correlation between maximum and minimum temperature, HWMId results with minimum temperature differ significantly from those with maximum temperature. Specifically, while HWMId$_{tx}$ is constantly increasing with time, HWMId$_{tn}$ displays a positive trend only from 1981 to the middle of the 1990s. Also, the number of stations affected by heat wave events of minimum temperature is pretty low (i.e., ~10 events in the period 1981-1985 and ~40 events in the

period 2011-2015 all over Africa).

Histograms in Fig. 4 show the temporal distribution of heat wave for each class of magnitude for both maximum and minimum temperature. The five year time window allows us to better visualize the evolution of heat waves. Results confirm previous findings of Fig. 2 and 3. For HWMId$_{tn} \geq 3$ there is a peak during 2006-2010, whereas for the other classes it is the period 2011-2015 that displays the higher number of occurrences. However, this might be due to the lack of 2015 data in the Northern

hemisphere (as explained in the Materials and Methods section) that slightly reduces the number of occurrences during the last five-year period.

The occurrence of heat waves of maximum temperature generally increased from 1981. For maximum temperature, the increase in heat wave events occurred for all the magnitude classes. For minimum temperature it is possible to observe a weak upwards trend only for heat waves with magnitude greater than three and six, with a peak in the period 1991-2000.

As for GSOD, Fig. 5 shows the HWMId$_{tx}$ of the *reanalysis* dataset for 5-year periods from 1981 to 2015. Unlike our finds for GSOD, HWMId$_{tn}$ - shown in Fig. 6 - do not differ significantly from HWMId$_{tx}$. Since the *reanalysis*-based HWMId is a gridded product, histograms in the bottom-right corners show the percentage of land area affected by heat wave (rather than the number of events) for each magnitude class. It is possible to distinguish an increase of heat wave intensity in the last 20 years. This is noticeable from the analysis of the histograms, where the maximum value of the percentage of land where heat waves occurred

rises from ~25% to ~60% per 5-year period.

*Reanalysis* spatial coverage throughout Africa is continuous, circumventing thus the limitations of GSOD, where the locations of the observing stations might lead to a difference between the increases in occurrences in all of African versus observations made at these specific stations. The major limitation of the *reanalysis* dataset is the fact that the HWMId metric is based on



daily minima and maxima, which is more accurately captured in the (spatially) sparse but high resolution stations than in *reanalysis* data.

Spatial patterns indicate an increase of HWMId$_{tx}$ during the last 20 years. The major hot spot of increases in HWMId frequency and magnitude are located in Northern Africa, ranging from Morocco to Egypt, and in Angola, Congo, South Sudan, Kenya

and Madagascar. Also HWMId$_{tn}$ shows an increase in frequency and magnitude of heat wave during the last 20 years, but it exhibits different spatial patterns. Generally it occurs - with less intensity- in the same zone hit by maximum temperatures, but it also affects other zones. This increasing in HWMId$_{tn}$ is noticeable across Angola, Congo, Zambia, Namibia and Botswana, while it is not present in Northern Africa. Opposite to what found for maximum temperature –which displays a positive trend from 1981 to 2015- minimum temperature has a positive trend in the period 1981-1996 and then remains steady. It also displays

a peak during 2006-2010.

Just as for GSOD, histograms in Fig. 7 show the temporal distribution of African land fraction affected by heat waves for each class of magnitude for both maximum and minimum temperature. Unlike what found for GSOD, the percentage of area interested by heat waves is increasing for both maximum and minimum temperature. Both HWMId$_{tx}$ and HWMId$_{tn}$ histograms display a peak during 2006-2010, just as for maximum temperature from GSOD. However, HWMId$_{tx}$ displays an ongoing

positive trend, whereas HWMId$_{tn}$ seems to remain steady from 1996 onwards.

An example of the performance of Reanlysis-derived Heat Wave is given in Fig. 8. The maps show the HWMId$_{tx}$ from both observations (i.e., GSOD) and *reanalysis* (i.e., ERA-INTERIM) for 1988, 1998, 2005, 2010, 2013, and 2014. The vast majority of the maps refers to the last 10 years, i.e. the period characterized by a significant increase in heat wave intensity and frequency. The remaining annual maps for the period 1981-2014 not shown in Fig. 8 are in the Annex.

For 1988, there is a good match between GSOD and *reanalysis* results across Northern Africa. Also 1998 heat waves across Western Africa and Morocco are well captured by both datasets. In regard to 2005, there is a good match throughout Northern Africa. During 2010 heat wave events are present in Northern Africa (i.e., Egypt, Libya, Niger and Burkina Faso). These events are similar in magnitude to those observed from the GSOD records. The 2012 heat wave events in Morocco and Algeria from GSOD are consistent with the *reanalysis*. Interestingly, this heat wave is well documented in both newspapers and

literature (e.g. Daily News Egypt, 2012 and Blunden and Arndt, 2013). Specifically, the heat wave and Ramadan were concomitant, leading to power outages and riots against governmental institutions and private corporations in Algeria. 2013 heat wave in Morocco from *reanalysis* are not intense as much as those observed from GSOD network. However, 2013 events in West Africa, Namibia, and in the zone between Libya and Egypt show good agreement. It is possible to observe consistency also during 2014 across Madagascar, Zimbabwe, and North Africa. Note that the 2014 heat wave in Madagascar is associated

with locust invasion and it is well documented in the press (Rzadkiewicz, 2014) and by the Food and Agriculture Organization (FAO) for an emergency report (FAO, 2016).

Although the qualitative character of the comparison due to the low GSOD station number, ERA-INTERIM shows good agreement with observations.



## 4 Conclusions

In this work we present the results of the application of the Heat Wave Magnitude Index daily (HWMId) in the assessment of climate change across Africa. Observation from GSOD and *reanalysis* from ERA-INTERIM datasets are used to identify heat waves and their temporal and spatial variability. GSOD observations are able to capture heat wave events at fine spatial scales,

but show a sparse coverage across Africa. Conversely, *reanalysis* dataset, although the coarse spatial resolution, displays a homogeneous coverage.

Results from heat wave analysis of maximum temperature ($HWMId_{tx}$) indicate an increase in intensity and frequency of extreme events. Specifically, from 1996 onwards it is possible to observe $HWMId_{tx}$ spread with the maximum presence during 2006-2015. Between 2006 and 2015 the frequency (*spatial coverage*) of extreme heat waves had increased to 24.5 observations

(*60.1% of land cover*) per year, as compared to 12.3 (*37.3% of land area*) per year in the period from 1981 to 2005 for GSOD stations (*reanalysis*).

Regarding heat waves of minimum temperature, GSOD-based $HWMId_{tn}$ shows a surge in in the middle of the 1990s. Conversely, the *reanalysis*-based $HWMId_{tn}$ shows a dramatic increase in extreme events. There is spatial coherence with the patterns relative to maximum temperature, even if heat waves also occurred in other regions. The highest positive trend in heat

wave events refers to the period 1981-1996 when the percentage of land cover affected by heat waves had doubled from ~20% in 1981 to ~40% in 1996.

Our results also show coherence between observation-based and *reanalysis*-based heat waves. However, this is just a qualitative comparison, and a correct evaluation could be possible only with a very dense (and well-established) meteorological network, such as those in Europe or the US.

The study also touches upon sociological aspects of the heat waves.Many events are well documented in the news, indicating that HWMId is able to capture events that are perceived as heat waves by a broader public.

There is agreement with the recent findings of the World Meteorological Organization (WMO): the years 2011-2015 have been the warmest five-year period on record (WMO, 2015) and heat waves of maximum temperature have increased both in severity and number accordingly.

Our work has direct relevance for both scientists and policy makers. Increasing numbers of heat waves may pose challenges on healthcare and on electric supply for residential cooling demands, among others. These implications argue for the importance of enhancing the density of hydro meteorological stations to provide the baseline data that will be essential: 1) for climate change adaptation, and 2) to improve the observation constrain of *reanalysis* products. Further applications include: 1) the employment of HWMId scheme to climatological models to quantify the increase in heat wave for the next decades; 2)

the wider and quantitative implications of African heat waves on health, crops, and finance; and 3) the analysis of teleconnections between the 2015/2016 El Nino event (Cesare, 2015) and heat waves in Eastern Africa.





**Acknowledgements**

Authors would like to thank the valuable support from JRC. This work has received funding from European Commission EuropeAid Co-operation Office under the grant agreement RALCEA. The data used in this manuscript can be obtained from: 1) Global Summary of the Day (GSOD) version 8, National Climatic Data Center (ftp://ftp.ncdc.noaa.gov/pub/data/gsod/);

5 and 2) ERA Interim, Daily produced by the European Centre for Medium-Range Weather Forecasts (ECMWF) (http://apps.ecmwf.int/datasets/data/interim-full-daily ). The authors also thank those responsible for the efforts on providing free tools such as R used in this work. The R packages were obtained from the Comprehensive R Archive Network (http://cran.r-project.org/ ) or R-forge (https://r-forge.r-project.org/ ). Heat Wave Magnitude Index daily has been calculated using the R library "extRemes" (Gilleland, 2015). Google data are registered trademarks of Google Inc., used with permission.

10 Maps are available upon request. The authors acknowledge Hugh Eva for his help for editing the paper.





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





Figure 1 Spatial distribution of temperature gauges used in this study.





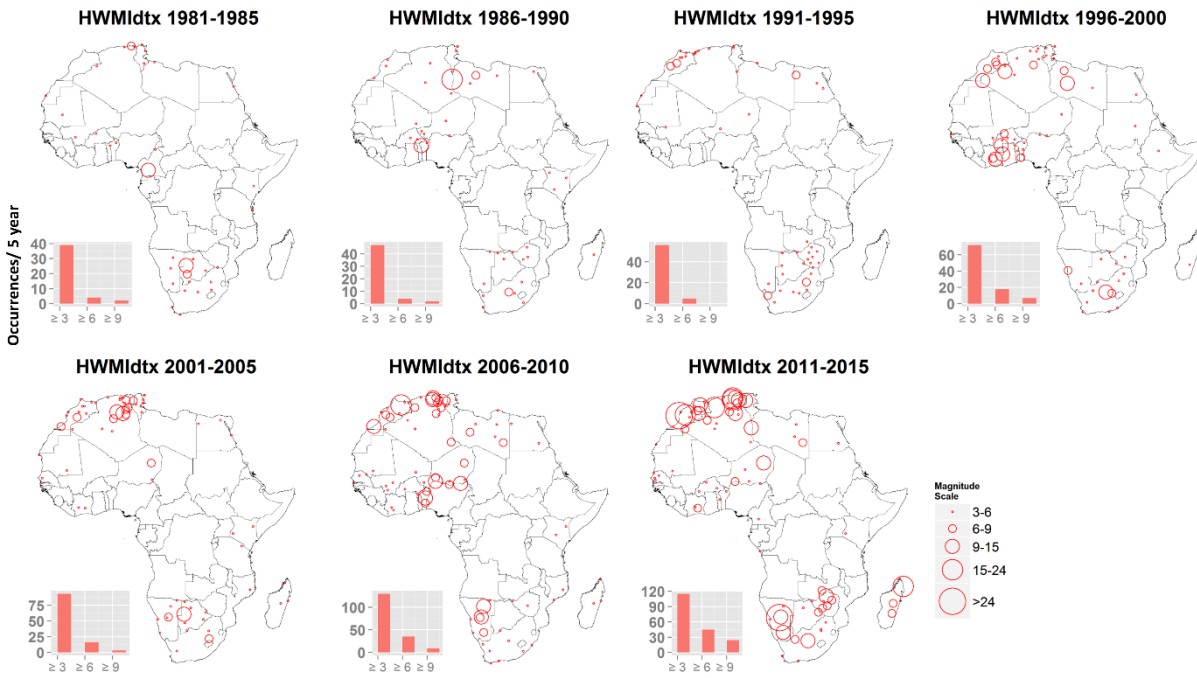

Figure 2 Magnitude of Heat Wave Index of maximum temperature (HWMId$_{tx}$) for 5-year periods of GSOD records from 1981 to 2015. Histograms in the bottom-right corners show the occurrence of the magnitude index classes for each 5-year period.





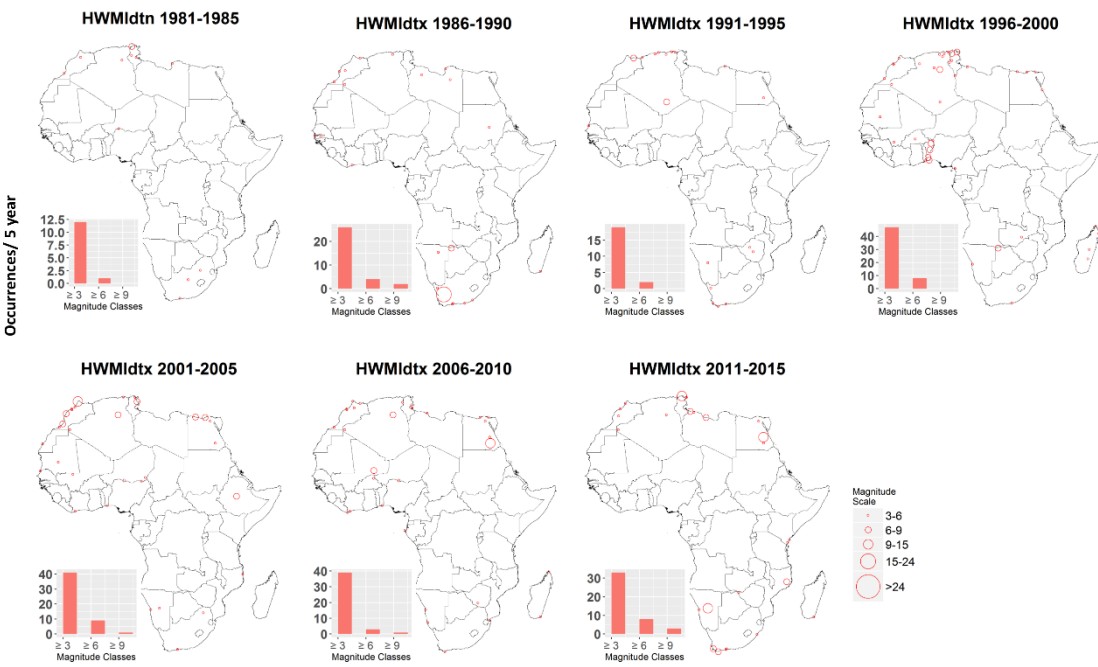

Figure 3 As Fig. 2 but for Heat Wave Magnitude of minimum temperature applied to minimum temperature (HWMIdtn).



Figure 4 Histogram of heat wave of maximum (TX, upper panel) and minimum (TN, bottom panel) temperature for 5-year
5    period of GSOD records during 1981-2015.




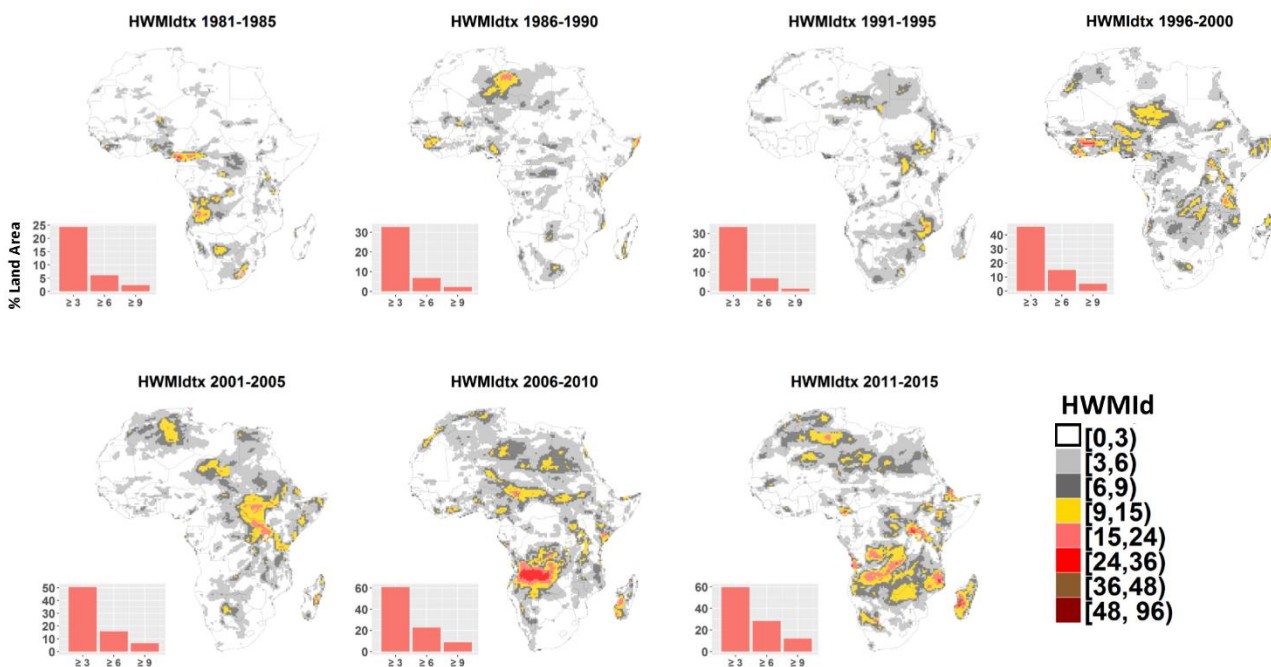

5  Figure 5 Magnitude of Heat Wave Index of maximum temperature (HWMId$_{tx}$) for 5-year periods of ERA-INTERIM dataset from 1981 to 2015. Histograms in the bottom-right corners show the spatial distribution of the magnitude index classes for each 5-year period.





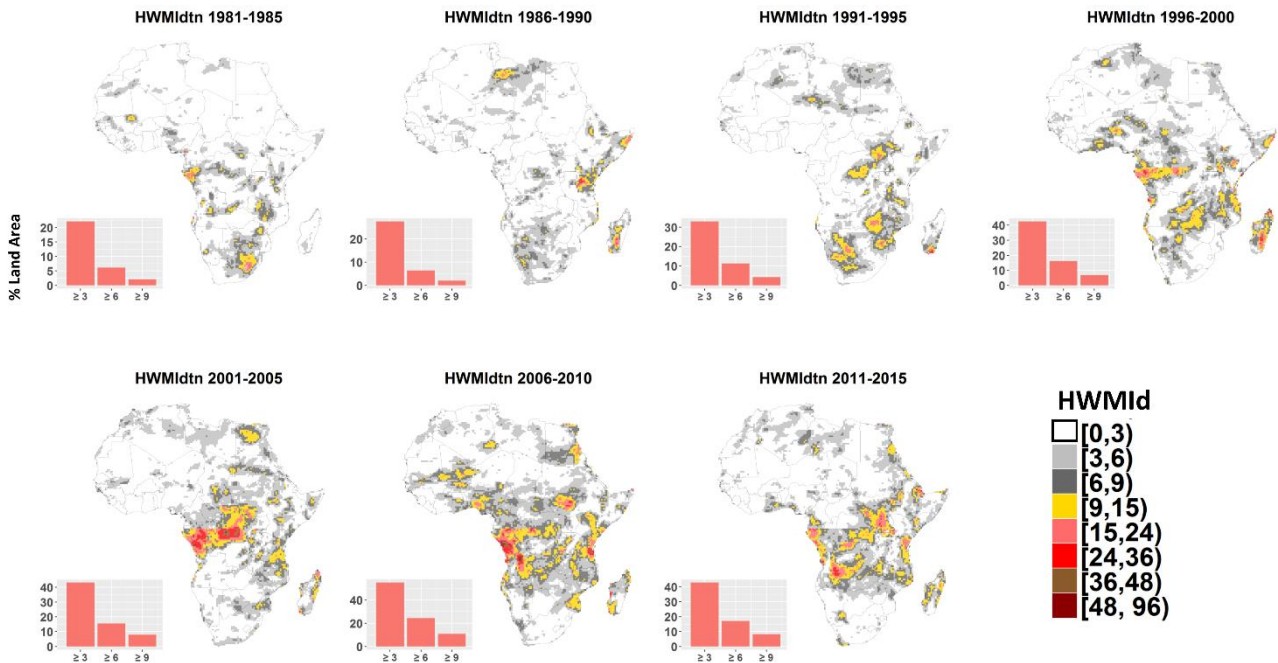

Figure 6 As Fig. 5 but for Heat Wave Index applied to minimum temperature (HWMId$_{tn}$).



Figure 7 Histogram of heat wave of maximum (TX, upper panel) and minimum (TN, bottom panel) temperature for 5-year
period of ERA-INTERIM dataset during 1981-2015.





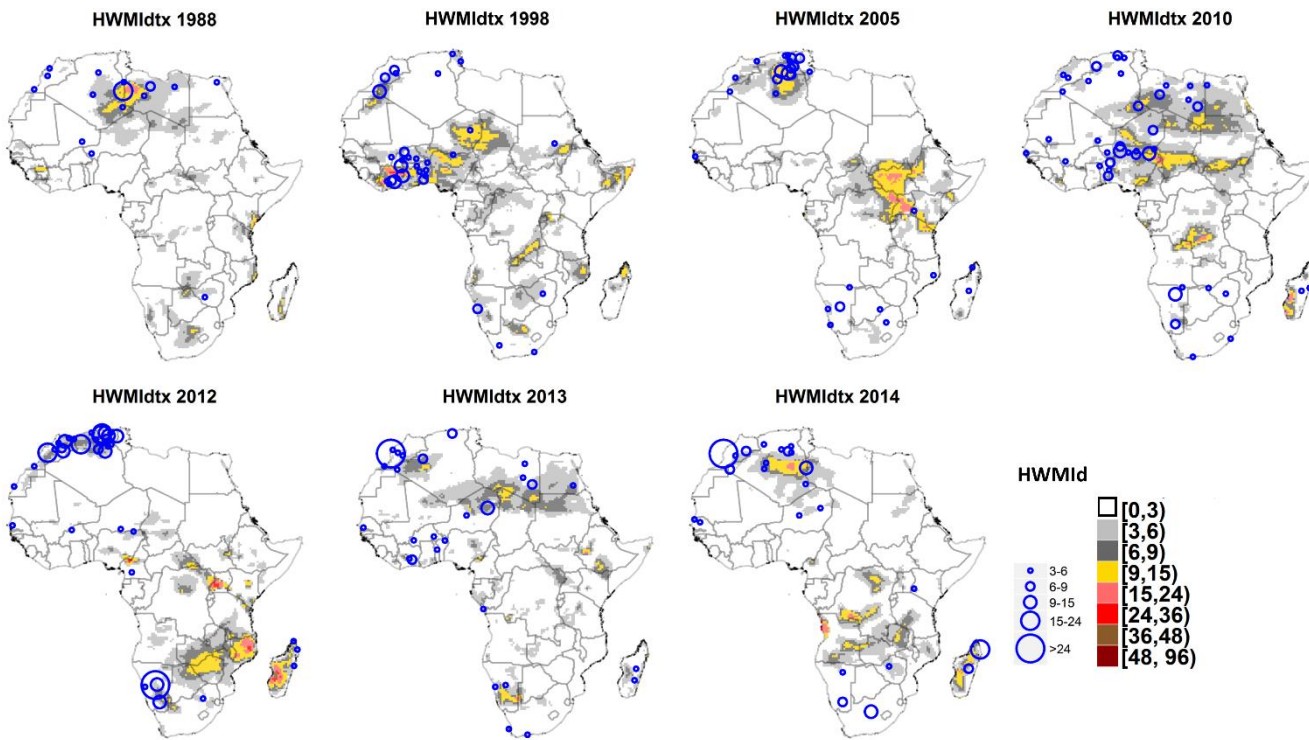

Figure 8 Comparison between HWMId$_{tx}$ from GSOD network and ERA-INTERIM dataset for 2005, 2010, 2012, 2013, and 2014.