# Peer review of "Heat waves in Africa 1981-2015, observations and reanalysis"

_Natural Hazards and Earth System Sciences, 2016_

## Referee Comment (RC1) · D. Lee (Referee) · 2 May 2016

**General comments**

This paper examines the presence of heat waves in Africa using two different data sources and a novel and robust descriptive metric. Although heat waves are of great importance for human society, the results are unfortunately not sufficiently embedded in a theoretical/practical context. Neither the materials nor the methods are described in sufficient detail to reproduce the results. The findings are salient but very cursory and explained only very curtly, using subjective evaluations to draw conclusions that are not necessarily made apparent to the reader. As such, the paper has the potential of making a relevant contribution to the research community, but I regrettably cannot recommend it for publication in this form.

**Specific comments**

**Overview**

All in all, the paper feels very data driven. This might not have been a problem if it were presented in a journal geared toward presenting data sets, and if the data were made available to the general research community, but neither is the case here. The subject matter and results are presented with a very minimalistic embedding in a wider context.

Reproducibility is another issue. The data produced in this study are not made available to the community - it would be a great improvement if this were published somewhere and referenced in the paper. Additionally, the materials used for the paper are not described with enough detail to reproduce the results. ERA-Interim is a large data set. What components were used to compute the maximum and minimum temperature for each day? The assimilation runs? Was absolute temperature at a certain time step or one of the derived temperature variables used? Multiple variables come into question and could arguably make sense, but as this is not explained the methodology is ambiguous.

Also, the conclusions presented in the paper are fairly subjective. In the absence of data, at least tables would be convenient. Instead, the reader is asked to "eyeball" figures. When discussing the findings, the authors mention trends. However, these are not quantified in any way. Quantification of the trends and their significance would provide a necessary measure of objectivity.

**Specifics**

This is not an exhaustive list.

*Page 1, line 10:* "The purpose of this article is to show the. . . regime of heat waves. . ." The paper does little to characterize the regime of heat waves. Intra-year variability receives no real mention, and all data is aggregated onto five yearly time scales.

[Figure]

*Page 2, lines 7-10:* It would be helpful if the relevance of the analyses were explained in greater detail. The authors mention that Africa is vulnerable, but they do not really describe how this study specifically sheds light on Africa's weather in a useful fashion. This is addressed in the conclusions, but in very broad strokes. More concreteness earlier would be better to motivate the reader to invest time in the paper.

*Page 3, lines 8-12:* If this paper were referring to data found elsewhere, it would not need to be very specific about how the data was computed. However, as the data was computed specifically for the paper, the methodology should be described precisely. The citations are okay, but it would still be nice to see the equations behind the HWMI. The text does not really explain what HMWI measures, and the equations would hopefully make that clear.

*Page 3, lines 20-23:* These are essential details that allow the reader to understand the meaning of all the numbers discussed later in the paper. What does the classification mean? In which events might one expect to see a heat wave classified as a certain number? Without this information it is very difficult to interpret the results. Sadly, the information is left out.

*Page 3, lines 24-28:* This leaves the tropics out of consideration. It's understandable that it's hard to touch on every issue when doing a study that compares tropics and two hemispheres but it would be nice to see that mentioned here.

*Page 3, line 29:* The time shift to compensate for the different seasonal regimes between hemispheres definitely does cause a temporal inconsistency. This is referred to later on in the paper as a possible reason why the northern and southern hemispheres have slightly different results. Therefore "might" is definitely incorrect. It would be better to address this inconsistency in the paper than to simply mention it.

*Page 4, lines 8-10:* It's not clear what this means.

*Page 4, line 13:* When describing a trend it would be helpful to include information

from a trend analysis - a visual analysis is very subjective.

*Page 4, line 19:* This isn't clear from the figure. Also, if the anomaly might be due to the differing length of the analyzed time series, surely it would be better to correct for this rather than to use it as a possible but unconfirmed reason for the inconsistency. Normalization or truncation of the data are two straightforward, but certainly not the only ways of doing this.

*Page 4, line 26:* A visual comparison of maps by putting them side by side is difficult. This information should be reduced - for example by providing histograms or change maps. Also, it is unclear what is being compared here. The intensity? Frequency? Spatial distribution? The trend over time?

*Page 4, lines 27-28:* What is the meaning of the gridded HMWI product? Five years are aggregated to a single map with only the spatial distribution of heat waves of a certain intensity. Does each map signify the highest HMWI computed at a given point over the five year period? Are frequently reoccurring heat events captured on these maps? I suspect not, and that would be interesting information. Regardless of whether or not this is the case, it's not clear and should be. Occurrence, intensity and frequency are all separate attributes of the heat wave phenomenon and would be a good addition to the paper. In general, referring to the percent of land affected by high HMWI is ambiguous. How long does a given HMWI class have to be reached in order for it to be considered affected over the five year period?

*Page 5, line 1:* The temporal resolution of the reanalysis data should be stated some-where. This would make it clearer for those interested in reproducing the paper's re-sults, particularly because several options are open.

*Page 5, lines 8-10:* Again, if a trend is described, it should be described objectively, i.e. with an actual trend analysis.

*Page 5, lines 13-14:* This is not clear from the figure.

*Page 5, lines 14-16:* This should be quantified.

*Page 5, lines 16-19:* This means little without more clarity, either through better explanations or quantification, or in the best case both.

*Page 5, lines 20-33:* This would have been a very interesting line of investigation to pursue, if some objectivity was provided. There are classic measures for verification that would shed much light on the ability of HMWI to describe heat events observed using other methods - one example is a confusion matrix, but many are available. The passage here falls far short of a true verification. Objectivity and quantifiable conclusions would be a large improvement.

*Page 6, line 20:* This simply is not the case, no sociological aspects are mentioned.

*Page 6, lines 22-23:* This would fit better in the introduction to catch the reader's interest and demonstrate the relevance of the study.

*Page 6, lines 27-31:* Many applications of the data produced in this study are mentioned here, but they are not mentioned in the paper as such. A theoretical section would be helpful and appropriate, otherwise the applications mentioned appear out of context.

**Technical corrections**

*General:* The description of the variables is nonuniform. Sometimes it's HMWId_tx, sometimes HMWIdtx, sometimes HMWId TX. This needs to be harmonized.

*General:* why is the word "reanalysis" always italicized?

*Page 1, line 15:* "... an increasing numbers. . ." -> "an increasing number"

*Page 1, lines 28-29:* This sentence is like many others in the document: grammatically correct but they feel backwards because they begin with a qualifier that describes the action in the sentence. You could consider turning these around.

*Page 3, lines 12, 14:* Here "according to the methodology described in..." might be clearer than "for further details..."

*Page 3, line 24:* here "in" would be more accurate than "between".

*Page 5, lines 16-17:* This sentence is unnecessary. Figures should be self-explaining where possible with any necessary descriptions in their captions. That way the reader is not required to page back and forth between the figures and the text in order to interpret them.

*Page 6, line 28:* " improve the observation constrain" - the wording is unclear.

*Figure 2:* The borders between the classes are ambiguous. 3-6, 6-9, etc. overlap. This should be e.g. 3-<6, 6-<9, if that if correct.

Also, the maps are difficult to compare. Most people have difficulties transferring the information from one map to another for comparison. If you are requiring the reader to overlay the maps in their mind, it is probably better to reduce the cognitive overhead by overlaying the maps in the figure so that they can more quickly recognize what you want to say. Admittedly, by the number of maps here that might be difficult but it's possible.

The histograms are easier to compare, but they are so small that the reader can only draw very basic conclusions from them. Since the histograms are repeated in figure 4, it would be better to leave them off and have a less crowded figure.

*Figure 3:* See figure 2. Additionally, a much more in-depth analysis would be possible - and it wouldn't need to be difficult! For example, scatterplots of the frequency of different classes of heat events for each station over time would be intuitive to interpret for most readers. They would also present a good opportunity to overlay trend lines produced with regressions, etc.

*Figure 4:* These plots could be reduced and it would make them much easier to interpret. It is more intuitive to put time on the x-axis than magnitude classes. Differentiating

the magnitude classes through the legend would also be more intuitive.

However, also here a scatterplot would be most helpful, in part because it would allow the visualization of trend lines and also because it would condense the information. Suggestion: Use the x-axis for time. Differentiate TX and TN by color. Use differing symbols for each time period and plot them next to each other with trend lines for each group.

*Figure 6:* See comments for figures 2, 3. Also, the plot's meaning is unclear. Does each grid point show the maximum HMWId reached in the given time period? What happens with multiple hits?

A more interesting analysis would be to show the likelihood that a given HMWI will be reached across the map, as well as change maps that show how this has changed time. The results could also be used to evaluate the likelihood of heat incidents occurring in major African population centers, etc.

*Figure 7:* The meaning of "% of Land Area" is unclear. Is this the percent of land area in which a certain HMWI class was reached? The percent that was reached in that class consistently in all years? Etc.

Also note that if you are using the native coordinate system of the ERA-Interim data, these numbers will not be correct, as the grid cells refer to unprojected coordinates whose areas change according to their proximity to the equator.

*Figure 8:* See other comments for other map figures. Again, a confusion matrix or something similar would be much better than a subjective "eyeballing" of the data.

---

## Referee Comment (RC2) · Anonymous Referee #2 · 8 May 2016

The paper describes an analysis of the heat waves in the last decades by using a synthetic index and based on both observations and reanalysis data (ERAInterim dataset). The paper is interesting and well written and the language is clear and correct. However, the description of the analysis lacks of details, and some Figure could be more clear. More specific comments follow: Pag. 1, Introduction: further references are needed, especially about the topics of the lack of data needed for the assessment of the heat waves, and for what concerns the climate change-related issues. Also, a reference on the Global Surface Summary of the Day is fundamental to better understand the type of data and the reliability and coverage of the dataset.

Pag. 1, line 31: which is exactly the density of the network in sensors/km2 in the region of interest of the analysis?

Pag. 2, lines 14-15: I would suggest to change the title of the Section 2 in "Data and methodology", the title of subsection 2.1 in "Data" and the title of Subsection 2.2 in "Methodology".

Pag. 2, subsection 2.1: Even if the correct references are present about the 4D-Var assimilation method and the topic of the use of reanalysis data, a specific reference about the ERAInterim dataset is absent, it should be added in the text. Pag. 3, subsection 2.2: Nevertheless the proper reference is present, a detailed description of the HWMId index should be included in the text. This is crucial, because, also if the authors state that this index resume different information about the single heat wave in a single scalar index, the exact definition is vital to understand what is it actually represented by the index itself, and to give a possible assessment of the validity of the analysis.

Pag, 3, lines 18-19: the authors state that the analysis was carried on for both the observational dataset (for each station separately) and for the reanalysis dataset on the whole African territory. Some clarifications are needed: how the values of ERAInterim were used? The index was computed separately for each cell (which was the spatial resolution of the reanalysis dataset? Temporal resolution?)? It appears so looking the Figures 5 and 6, but it should be stated explicitly in the text. If for ERAInterim the whole African territory was used, how exactly the analyses carried on both dataset can be considered comparable? This is an important issue given the low spatial density of the GSOD sensors and, mainly, its inhomogeneous spatial distribution, and it should be clarified.

Pag. 4, lines 4-15: the Figure 2 and 3 illustrate the occurrence of heat waves for periods of 5 years. The authors state that an increasing trend is observable, as clear for Figure 4. I would suggest to use the same vertical limits for all the histograms in order to make them more comparable, or to eliminate them, since they are anyway repeated in Figure 4.

Pag. 6, Conclusions: the authors state that the analysis is of broad interest and suggest

to extend it to the dataset produced by climate models in climate change scenarios. This is interesting, however, it should be stressed that even reanalysis data, such the ones employed in this work, are subject to errors with respect to the observations, where the latest are available. This work itself shows an example with the incoherence between the results of GSOD and ERAInterim with respect to the night temperatures. Thus, at least some considerations about the uncertainty present in reanalysis data and, even more, in future scenarios, should be added, in order to clarify the scope of the analysis-

Considering the above-reported observations, I suggest minor revision before publication.

---

## Author Comment (AC1) · 15 Jun 2016

**# General comments**

*This paper examines the presence of heat waves in Africa using two different data sources and a novel and robust descriptive metric. Although heat waves are of great importance for human society, the results are unfortunately not sufficiently embedded in a theoretical/practical context. Neither the materials nor the methods are described in sufficient detail to reproduce the results. The findings are salient but very cursory and explained only very curtly, using subjective evaluations to draw conclusions that are not necessarily made apparent to the reader. As such, the paper has the potential of making a relevant contribution to the research community, but I regrettably cannot recommend it for publication in this form.*

**# Specific comments ##**

*Overview*

1) *All in all, the paper feels very data driven. This might not have been a problem if it were presented in a journal geared toward presenting data sets, and if the data were made available to the general research community, but neither is the case here. The subject matter and results are presented with a very minimalistic embedding in a wider context.*

We agree with the above reviewer comment that results are presented with a minimalistic embedding. The manuscript has been rewritten in most places accordingly.

2) *Reproducibility is another issue. The data produced in this study are not made available to the community - it would be a great improvement if this were published somewhere and referenced in the paper.*

Thank you for this comment. In the previous manuscript version we forgot to remark that both ERA-Interim and GSOD data are of public domain. (see http://apps.ecmwf.int/datasets/data/interim-full-daily/levtype=sfc/ and http://www7.ncdc.noaa.gov/CDO/cdoselect.cmd?datasetabbv=GSOD&countryabbv=&georegionabbv= )

Moreover the HWMId function has been published on R CRAN in a recent update of the packages extRemes (Gillelard 2011). Reproducing HWMId data is very simple by applying the HWMId R function to the above free available datasets.

3) *Additionally, the materials used for the paper are not described with enough detail to reproduce the results. ERA-Interim is a large data set. What components were used to compute the maximum and minimum temperature for each day? The assimilation runs? Was absolute temperature at a certain time step or one of the derived temperature variables used? Multiple variables come into question and could arguably make sense, but as this is not explained the methodology is ambiguous.*

Thanks for this fair comment. We have added a thorough description of the ERA-INTERIM database. We have rephrased the text such as:

**"ERA-INTERIM (Berrisford et al., 2011; Dee et al., 2011) is the dataset of reanalysis providing hydro meteorological variables such as maximum and minimum temperature, evaporation, snowfall, runoff and precipitation across land at various temporal scales. Reanalysis has been increasingly used to address a variety of climate-change issues and has by now become an important method in climate change research (Fan and van den Dool, 2004; Marshall and Harangozo, 2000; Uppala et al., 2005). ERA-INTERIM is a reanalysis product of the European Centre for Medium-Range Weather Forecasts (ECMWF) available from 1979 and continuously updated in real time. The data assimilation system used to produce ERA-INTERIM is based on a 4-dimensional variational scheme (4D-Var) with a 12-hour analysis window (for further information on 4D-VAR see Courtier et al., 1994). In this study we have used 2-meter daily maximum and minimum temperature at a spatial resolution of 0.75 degrees. The reanalysis dataset used in this study has a spatial resolution of 0.75° (i.e., approximately 80 km at the equator), and a temporal resolution of one day (i.e., a time step of 24 hours). The variables of interest are daily maximum and minimum temperature, ranging from 1981 to 2015."**

4) *Also, the conclusions presented in the paper are fairly subjective. In the absence of data, at least tables would be convenient. Instead, the reader is asked to "eyeball" figures.*

Thanks for this fair comment. We have reduced the number of figures and we added tables and scatterplots in the manuscript and in the Annex.

5) *When discussing the findings, the authors mention trends. However, these are not quantified in any way. Quantification of the trends and their significance would provide a necessary measure of objectivity.*

We do agree with this comment. Trend values (and the relative statistical significance of the coefficients) have been quantified and reported in the text. To this end we have modified Figure 4 and 7.

**Specifics This is not an exhaustive list.**

1) *Page 1, line 10:* *"The purpose of this article is to show the regime of heat waves. . ." The paper does little to characterize the regime of heat waves. Intra-year variability receives no real mention, and all data is aggregated onto five yearly time scales.*

Thank you, we have now updated the manuscript showing the occurrence of heart wave at annual scale

2) *Page 2, lines 7-10:* *It would be helpful if the relevance of the analyses were explained in greater detail. The authors mention that Africa is vulnerable, but they do not really describe how this study specifically sheds light on Africa's weather in a useful fashion. This is addressed in the conclusions, but in very broad strokes. More concreteness earlier would be better to motivate the reader to invest time in the paper. *

We appreciate this fair comment. According to the reviewer's comment and we updated the text.

"

**Africa is considered one of the most vulnerable regions to weather and climate variability (Solomon et al., 2007): extreme events such as heat waves are doing increasing damage to health, water, energy and food systems. According to Albrecht (2014), climate change will increase its pressure in North Africa. All capital cities in the region could face many more exceptionally hot days each year and - compared to the rest of the world - North Africa will disproportionally suffer from heat wave events.**

**Besides, recent findings of the World Meteorological Organization (WMO) indicate that the years 2011-2015 have been the warmest five-year period on record (WMO, 2015) and heat waves of maximum temperature have increased both in severity and number accordingly.**

**However, although its vulnerability, the real distribution of African heat waves is still poorly understood and the task of developing a reliable assessment is further complicated by the lack of accurate baseline data on current climate (UNECA, 2011). Thus, variability and changes in extreme temperature regimes present a considerable challenge since different aspects - both spatially and temporally - are still lacking.**

**…….**

**Considering both its wide geographic scope and spatial resolution, the study represents an important step towards the assessment of heat wave frequency in the last three decades. The availability of such information is paramount. The more reliable the assessment of heatwaves is, the better African countries will be equipped to strengthen their coping capacities.***"*

3) *Page 3, lines 8-12:* *If this paper were referring to data found elsewhere, it would not need to be very specific about how the data was computed. However, as the data was computed specifically for the paper, the methodology should be described precisely. The citations are okay, but it would still be nice to see the equations behind the HWMI. The text does not really explain what HMWI measures, and the equations would hopefully make that clear.* *

We have modified the text and we have described thoroughly the HWMId computation:

"**The HWMId is defined as the maximum magnitude of the heatwaves in a year. Specifically, a heatwave is defined as a period ≥ 3 consecutive days with maximum temperature above a daily threshold calculated for a 30-year long reference period. The threshold is defined as the 90th percentile of daily maxima temperature, centered on a 31 day window.**

**The inter quartile range (IQR, i.e, the difference between the 25th and 75th percentiles of daily maximum temperatures) is used as the heatwave magnitude unit, since it represents a non-parametric measure of the variability. If a day of a heatwave has a temperature value equal to IQR, its corresponding magnitude value will be equal to one. According to this definition, if the magnitude on the day d is 3, it means that the temperature anomaly on the day d is 3 times the IQR. The HWMId has been already used to evaluate future impacts of heat waves in Africa until 2100 under different representative concentration pathways scenarios (Russo et al., 2016).**"

4) *Page 3, lines 20-23:* *These are essential details that allow the reader to understand the meaning of all the numbers discussed later in the paper. What does the classification mean? In which events might one expect to see a heat wave classified as a certain number? Without this information it is very difficult to interpret the results. Sadly, the information is left out.* *

Thanks for this comment, we have modified the text and we have briefly explained how to interpret the classification (see previous point).

5) *Page 3, lines 24-28:* *This leaves the tropics out of consideration. It's understandable that it's hard to touch on every issue when doing a study that compares tropics and two hemispheres but it would be nice to see that mentioned here.* *

The reviewer comment is correct. We modified the text by:

"**Note that this scheme leaves the tropics out of consideration.**"

6) *Page 3, line 29:* The time shift to compensate for the different seasonal regimes between hemispheres definitely does cause a temporal inconsistency. This is referred to later on in the paper as a possible reason why the northern and southern hemispheres have slightly different results. Therefore "might" is definitely incorrect. It would be better to address this inconsistency in the paper than to simply mention it. *

Ok, we remove the "might".

7) *Page 4, lines 8-10:* It's not clear what this means. *

In order to show that we have found the same results as in Fontaine et al., (2013), we have rephrased the text as follow :

**"Such increase of heat wave frequency corresponds with the findings of Fontaine et al. (2013): the occurrence of heat waves has clearly increased after 1996."**

8) *Page 4, line 13:* When describing a trend it would be helpful to include information from a trend analysis - a visual analysis is very subjective. *

Thanks for this comment. We have decided to compute trends (and linear regressions) of the occurrence of heat wave and to show them instead of the histograms (moved to the Annex). We have quantified trends with a linear regression with alpha = 0.01. Text has been modified accordingly.

9) *Page 4, line 19:* This isn't clear from the figure. Also, if the anomaly might be due to the differing length of the analyzed time series, surely it would be better to correct for this rather than to use it as a possible but unconfirmed reason for the inconsistency. Normalization or truncation of the data are two straightforward, but certainly not the only ways of doing this. *

We have removed this discussion from the manuscript.

10) *Page 4, line 26:\* A visual comparison of maps by putting them side by side is difficult. This information should be reduced - for example by providing histograms or change maps. Also, it is unclear what is being compared here. The intensity? Frequency? Spatial distribution? The trend over time? \**

We meant "the temporal evolution of HWMId$_{tn}$" and we rephrased the text accordingly.

11) *Page 4, lines 27-28:\* What is the meaning of the gridded HMWI product? Five years are aggregated to a single map with only the spatial distribution of heat waves of a certain intensity. Does each map signify the highest HMWI computed at a given point over the five year period? Are frequently reoccurring heat events captured on these maps? I suspect not, and that would be interesting information. Regardless of whether or not this is the case, it's not clear and should be. Occurrence, intensity and frequency are all separate attributes of the heat wave phenomenon and would be a good addition to the paper. In general, referring to the percent of land affected by high HMWI is ambiguous. How long does a given HMWI class have to be reached in order for it to be considered affected over the five year period? \**

Thanks for this comment. We modified the text and the figure accordingly.

**"Each pixel of the map represents the highest HWMId over the 5-year period. The 5-year temporal aggregation on the one hand simplifies the visualization, on the other hand hinders the detection of frequently reoccurring events. However, these event occurring at annual scale are analysed in Fig.7. In the case of ERA-INTERIM, instead of counting the number of occurrences, we have estimated the spatial extent of heat waves as the land area fraction exceeding a fixed HWMId value. The area fraction is expressed in percentage. Reanalysis spatial coverage throughout Africa is continuous, circumventing thus the limitations of GSOD, where the locations of the observing stations might lead to a difference between the increases in occurrences in all of African versus observations made at these specific stations. The major limitation of the reanalysis is that uncertainties are difficult to understand and quantify (Simmons et al., 2010).** "

12) *Page 5, line 1:\* The temporal resolution of the reanalysis data should be stated somewhere. This would make it clearer for those interested in reproducing the paper's results, particularly because several options are open. \**

The temporal resolution of the data is daily see comment above.

13) *Page 5, lines 8-10:* *Again, if a trend is described, it should be described objectively, i.e. with an actual trend analysis. **

We have carried out a trend analysis for both GSOD and ERA-INTERIM.

**"Figure 4 shows the occurrence of HWMId greater than a given magnitude level (i.e. HWMId ≥ 3, 6, 9, 15) for maximum and minimum temperature, respectively. Frequently reoccurring heat events are captured on these plots."**

14) *Page 5, lines 13-14:* *This is not clear from the figure.*

We have removed this sentence.

15) *Page 5, lines 14-16:* *This should be quantified. **

We have quantified the linear regression in Fig. 7

16) *Page 5, lines 16-19:* *This means little without more clarity, either through better explanations or quantification, or in the best case both. **

We agree with this comment. We have modified the text such as:

**"A visual comparison of the Heat Wave detection from observations (i.e., GSOD) and reanalysis (i.e., ERA-INTERIM) is given in Fig. 8. The maps show the HWMId$_{tx}$ in 1988, 1998, 2005, 2012 and 2014 as detected by GSOD network (blue circles) and ERA-INTERIM (gridded maps)."**

17) *Page 5, lines 20-33:* *This would have been a very interesting line of investigation to pursue, if some objectivity was provided. There are classic measures for verification that would shed much light on the ability of HMWI to describe heat events observed using other methods - one example is a confusion matrix, but many are available. The passage here falls far short of a true verification. Objectivity and quantifiable conclusions would be a large improvement. **

We added a confusion matrix (in the manuscript) and the scatterplot (in the annex). Text has been modified such as:

"**A quantitative comparison has been carried out by computing the confusion matrix of heat wave detection from observation and reanalysis. Table 1 and 2 shows the confusion matrices for the entire period 1981-2015 for maximum and minimum temperature, respectively. Heatwaves have been classified into four classes, i.e. HWMId ≤ 1, 1 < HWMId ≤ 3, 3 < HWMId ≤ 6 and HWMId > 6. For the sake of simplicity we omitted the number of events not classified as heat waves (i.e., HWMId ≤ 1) by both observation and reanalysis.**

**The vast majority of the elements of the matrices are not on the top-left to bottom-right diagonal, i.e. the correct classification. Besides, we can observe a "decay" of the number of events correctly classified when the magnitude level increases. This is also due to the lower number of intense heat waves compared to the moderate ones.**

**The off-diagonal elements represent classification errors, i.e. the number of heat waves that ended up in another class during GOSD and ERA-INTERIM classification. For both maximum and minimum temperature we can observe that ERA-INTERIM often underestimates GSOD-based heat waves.**

**Overall, the values of accuracy of classification for maximum and minimum temperature are 0.58 and 0.64, respectively. Note that these values are highly influenced by the correct detection of HWMId ≤ 1 which represent the vast majority of the area. The sensitivity, i.e. an indicator of the performance of the classifier, indicates that HWMId ≤ 3 are easier to be detected than higher classes by both databases. The scatterplots of observations versus reanalysis are shown in Fig. S3 of the Annex.**"

18) *Page 6, line 20:* This simply is not the case, no sociological aspects are mentioned. *

Thanks for this comment. The sentence has been removed.

19) *Page 6, lines 22-23:* This would fit better in the introduction to catch the reader's interest and demonstrate the relevance of the study. *

Done

20) *Page 6, lines 27-31:* Many applications of the data produced in this study are mentioned here, but they are not mentioned in the paper as such. A theoretical section would be helpful and appropriate, otherwise the applications mentioned appear out of context.*

Instead of adding a theoretical section, we have added in the methodology section further applications of the HWMId over Africa:

**"The HWMId has been already used to evaluate future impacts of heat waves in Africa until 2100 under different representative concentration pathways scenarios (Russo et al., 2016)."**

**# Technical corrections**

1)      *General:* The description of the variables is nonuniform. Sometimes it's HMWId_tx, sometimes HMWIdtx, sometimes HMWId TX. This needs to be harmonized.

Done

2)      *General:* why is the word "reanalysis" always italicized? *

The word reanalysis is not italicized anymore.

3)      *Page 1, line 15:* "... an increasing numbers. . ." -> "an increasing number" *

Done

4)      *Page 1, lines 28-29:* This sentence is like many others in the document: grammatically correct but they feel backwards because they begin with a qualifier that describes the action in the sentence. You could consider turning these around.

*Done*

5)      *Page 3, lines 12, 14:* Here "according to the methodology described in. . ." might be clearer than "for further details. . ." *

Done

6)      Page 3, line 24:* here "in" would be more accurate than "between". *

We used the word "between" to: 1) avoid repetitions (there are 5 in!) and 2) to highlight the fact that heatwaves might fall between these months thus hindering HWMId retrieval

7)      Page 5, lines 16-17:* This sentence is unnecessary. Figures should be self-explaining where possible with any necessary descriptions in their captions. That way the reader is not required to page back and forth between the figures and the text in order to interpret them. *

Done

8)      Page 6, line 28:* " improve the observation constrain" - the wording is unclear. *

We meant that in situ observations provide the information for up-to-date global analyses and climate reanalyses of the atmosphere, ocean and land surface. We rephrased the text such as "**Reduce the uncertainties**"

9)      Figure 2:* The borders between the classes are ambiguous. 3-6, 6-9, etc. overlap. This should be e.g. 3- Also, the maps are difficult to compare. Most people have difficulties transferring the information from one map to another for comparison. If you are requiring the reader to overlay the maps in their mind, it is probably better to reduce the cognitive overhead by overlaying the maps in the figure so that they can more quickly recognize what you want to say. Admittedly, by the number of maps here that might be difficult but it's possible. The histograms are easier to compare, but they are so small that the reader can only draw very basic conclusions from them. Since the histograms are repeated in figure 4, it would be better to leave them off and have a less crowded figure. *

We removed the histograms as suggested.

10)      Figure 3:* See figure 2. Additionally, a much more in-depth analysis would be possible - and it wouldn't need to be difficult! For example, scatterplots of the frequency of different classes of heat events for each station over time would be intuitive to interpret for most readers. They would also present a good opportunity to overlay trend lines produced with regressions, etc. *

We have added the time series of the occurrence of heatwave for different classes of events (see Fig. 4 and the new text).

11)      Figure 4:* These plots could be reduced and it would make them much easier to interpret. It is more intuitive to put time on the x-axis than magnitude classes. Differentiating the magnitude classes through the legend would also be more intuitive. However, also here a scatterplot would be most helpful, in part because it would allow the visualization of trend lines and also because it would condense the information. Suggestion: Use the x-axis for time. Differentiate TX and TN by color. Use differing symbols for each time period and plot them next to each other with trend lines for each group. *

Figure 4 has been moved to the annex. We have plotted the trend of occurrences of HWMId greater than a given threshold instead.

12)      Figure 6:* See comments for figures 2, 3. Also, the plot's meaning is unclear. Does each grid point show the maximum HMWId reached in the given time period? What happens with multiple hits? A more interesting analysis would be to show the likelihood that a given HMWI will be reached across the map, as well as change maps that show how this has changed time. The

*results could also be used to evaluate the likelihood of heat incidents occurring in major African population centers, etc. ***

**"Each pixel of the map represents the highest HWMId over the 5-year period. The 5-year temporal aggregation on the one hand simplifies the visualization, on the other hand hinders the detection of frequently reoccurring events. However, these event occurring at annual scale are analysed in Fig.7. In the case of ERA-INTERIM, instead of counting the number of occurrences, we have estimated the spatial extent of heat waves as the land area fraction exceeding a fixed HWMId value. The area fraction is expressed in percentage. Reanalysis spatial coverage throughout Africa is continuous, circumventing thus the limitations of GSOD, where the locations of the observing stations might lead to a difference between the increases in occurrences in all of African versus observations made at these specific stations. The major limitation of the reanalysis is that uncertainties are difficult to understand and quantify (Simmons et al., 2010).**

**"**

13)      Regarding the population, we added a population vs heatwave analysis for two different years. We added Fig. 9 and modified the text such as:

**"Figure 9 shows the density plot of population affected by heat waves detected by reanalysis in 2011 and 2012. Population count refers to the LandScan database (Bright et al., 2012) and it has been resampled to the ERA-INTERIM cell size, i.e. 0.75° x 0.75°. Note that the cell area of the gridded ERA-INTERIM product varies depending on the longitude. As expected, low-intensity heatwaves (i.e. HWMId ≤ 3) are more frequent across highly populated areas, but interestingly the highest events might affect the vast majority of African population. Both 2011 and 2012 indicates that heat waves generally wreak havoc on populated areas instead of the deserted ones. The fact that two different years present similar patterns stress the vulnerability of the African region and the importance of the heat wave assessment and prediction. "**

*14)      Figure 7:* The meaning of "% of Land Area" is unclear. Is this the percent of land area in which a certain HMWI class was reached? The percent that was reached in that class consistently in all years? Etc. Also note that if you are using the native coordinate system of the ERA-Interim data, these numbers will not be correct, as the grid cells refer to unprojected coordinates whose areas change according to their proximity to the equator. **

Meaning of % of land area: see point 12.

We acknowledge the limitation of the un projected coordinates (see previous point) but we do believe that this does not affect significantly the results.

*14)      Figure 8:* See other comments for other map figures. Again, a confusion matrix or something similar would be much better than a subjective "eyeballing" of the data.*

We added two confusion matrices (Table 1), a scatterplot and we reduced the number of map in Fig. 8 to avoid the eyeballing effect.

**Reviewer #2**

*The paper describes an analysis of the heat waves in the last decades by using a synthetic index and based on both observations and reanalysis data (ERAInterim dataset). The paper is interesting and well written and the language is clear and correct.*

*However, the description of the analysis lacks of details, and some Figure could be more clear. More specific comments follow:*

   1) *Pag. 1, Introduction: further references are needed, especially about the topics of the lack of data needed for the assessment of the heat waves, and for what concerns the climate change-related issues.*

Thanks for this fair comment. We agree with the reviewer and we added references and explanation to point out that the task of developing reliable assessment of climate change in Africa is further complicated by the lack of accurate baseline data on current climate. We modified the introduction accordingly.

   2) *Also, a reference on the Global Surface Summary of the Day is fundamental to better understand the type of data and the reliability and coverage of the dataset.*

We added a reference for GSOD: "**GSOD records underwent extensive automated quality controls to eliminate many of the random errors found in the original data (Further details on the GSOD data can be obtained from the website http://www.climate.gov/global-summary-day-gsod.).**"

   3) *Pag. 1, line 31: which is exactly the density of the network in sensors/km2 in the region of interest of the analysis?*

The spatial distribution is very patchy, however the density is 8.603574e-06 sensors/km2.

4) *Pag. 2, lines 14-15: I would suggest to change the title of the Section 2 in "Data and methodology", the title of subsection 2.1 in "Data" and the title of Subsection 2.2 in "Methodology".*

We have modified the titles.

5) *Pag. 2, subsection 2.1: Even if the correct references are present about the 4D-Var assimilation method and the topic of the use of reanalysis data, a specific reference about the ERAInterim dataset is absent, it should be added in the text.*

Ok, we added a specific reference

6) *Pag. 3, subsection 2.2: Nevertheless the proper reference is present, a detailed description of the HWMId index should be included in the text. This is crucial, because, also if the authors state that this index resume different information about the single heat wave in a single scalar index, the exact definition is vital to understand what is it actually represented by the index itself, and to give a possible assessment of the validity of the analysis.*

Thanks for this fair comment. We added a detailed description of HWMId is needed as reported in a comment above.

***"More precisely, HWMI has some problems in assigning magnitude to very high temperatures in a changing climate, thus resulting in an underestimation of extreme events (see Russo et al 2015). The HWMId is defined as the maximum magnitude of the heatwaves in a year. Specifically, a heatwave is defined as a period ≥ 3 consecutive days with maximum temperature above a daily threshold calculated for a 30-year long reference period. The threshold is defined as the 90th percentile of daily maxima temperature, centered on a 31 day window. The inter quartile range (IQR, i.e, the difference between the 25th and 75th percentiles of the daily maxima temperature) is used as the heatwave magnitude unit, since it represents a non-parametric measure of the variability. If a day of a heatwave has a temperature value equal to IQR, its corresponding magnitude value will be equal to one. According to this definition, if the magnitude on the day d is 3, it means that the temperature anomaly on the day d is 3 times the IQR. The HWMId has been already used to evaluate future impacts of heat waves in Africa until 2100 under different representative concentration pathways scenarios (Russo et al., 2016)."***

7) *Pag, 3, lines 18-19: the authors state that the analysis was carried on for both the observational dataset (for each station separately) and for the reanalysis dataset on the whole African territory. Some clarifications are needed: how the values of ERAInterim were used? The index was computed*

*separately for each cell (which was the spatial resolution of the reanalysis dataset? Temporal resolution?)? It appears so looking the Figures 5 and 6, but it should be stated explicitly in the text.*

We agree with the review. We have modified the text such as:

**"HWMId has been computed on annual basis:**

**1) for each GSOD station, and**

**2) for the entire spatial domain of the daily ERA-INTERIM maxima and minima temperatures dataset across Africa. Specifically, the index has been computed separately for each cell at ~80 km spatial resolution.**

**"**

And

**"Each pixel of the map represents the highest HWMId over the 5-year period. The 5-year temporal aggregation on the one hand simplifies the visualization, on the other hand hinders the detection of frequently reoccurring events. However, these event occurring at annual scale are analysed in Fig.7. In the case of ERA-INTERIM, instead of counting the number of occurrences, we have estimated the spatial extent of heat waves as the land area fraction exceeding a fixed HWMId value. The area fraction is expressed in percentage. Reanalysis spatial coverage throughout Africa is continuous, circumventing thus the limitations of GSOD, where the locations of the observing stations might lead to a difference between the increases in occurrences in all of African versus observations made at these specific stations. The major limitation of the reanalysis is that uncertainties are difficult to understand and quantify (Simmons et al., 2010).**

**"**

8) *If for ERAInterim the whole African territory was used, how exactly the analyses carried on both dataset can be considered comparable? This is an important issue given the low spatial density of the GSOD sensors and, mainly, its inhomogeneous spatial distribution, and it should be clarified.*

9) *Pag. 4, lines 4-15: the Figure 2 and 3 illustrate the occurrence of heat waves for periods of 5 years. The authors state that an increasing trend is observable, as clear for Figure 4. I would suggest to use the same vertical limits for all the histograms in order to make them more comparable, or to eliminate them, since they are anyway repeated in Figure 4.*

We have decided to eliminate them.

10) *Pag. 6, Conclusions: the authors state that the analysis is of broad interest and suggest to extend it to the dataset produced by climate models in climate change scenarios. This is interesting, however, it should be stressed that even reanalysis data, such the ones employed in this work, are subject to errors with respect to the observations, where the latest are available.*

We have modified the text to acknowledge this limitation:

"**. The major limitation of the reanalysis is that uncertainties are difficult to understand and quantify (Simmons et al., 2010).**"

And

"**Conversely, reanalysis dataset, although the coarse spatial resolution and the uncertainties, displays a homogeneous coverage.**

"

11) *This work itself shows an example with the incoherence between the results of GSOD and ERAInterim with respect to the night temperatures. Thus, at least some considerations about the uncertainty present in reanalysis data and, even more, in future scenarios, should be added, in order to clarify the scope of the analysis.*

We agree with the reviewer. We modified the conclusions adding:

"**Minimum temperature exhibits incoherence between the results of GSOD and ERA-INTERIM. GSOD-based HWMId$_{tn}$ shows a positive trend only for stations experiencing HWMId values greater than three. Conversely, the reanalysis-based HWMId$_{tn}$ shows a dramatic increase in extreme events. This rise, albeit minor, is comparable with that pertaining to maximum temperature.**"

Uncertainties have been already acknowledged.

New Figures and Table:

[Figure]

Figure 1 Heat Wave Magnitude Index daily of maximum temperature (HWMId$_{tx}$) for 5-year periods of GSOD records from 1981 to 2015.

[Figure]

Figure 3 As Fig. 2 but applied to minimum temperature (HWMId$_{tn}$).

[Figure]

Figure 4 Annual distribution of events exceeding four different thresholds (i.e., HWMId ≥ 3, 6, 9, 15) for maximum (top panel) and minimum (bottom panel) temperature. The blue line represents the statistically significant ($\alpha = 0.01$) linear regression.

[Figure]

Figure 5 Heat Wave Magnitude Index daily of maximum temperature (HWMId$_{tx}$) for 5-year periods of ERA-INTERIM dataset from 1981 to 2015.

[Figure]

Figure 6 As Fig. 5 but applied to minimum temperature (HWMId$_{tn}$).

[Figure]

Figure 7 Annual distribution of events exceeding four different thresholds (i.e., HWMId ≥ 3, 6, 9, 15) for maximum (top panel) and minimum (bottom panel) temperature. Since the reanalysis-based HWMId is a gridded product, plots show the percentage of land area affected by heat waves rather than the number of events.

[Figure]

Figure 8 Comparison between HWMId$_{tx}$ from GSOD network and ERA-INTERIM dataset for 1988, 1998, 2005, 2012 and 2014.

[Figure]

Figure 9 Density plot of population affected by heat waves of maximum temperature detected by reanalysis in 2011 (top panel) and 2012 (bottom panel). The y-axis refers to the population count per cell, where the cell is the 0.75° x 0.75° ERA-INTERIM pixel.

|  |  | ERA-INTERIM | | | |
|---|---|---|---|---|---|
|  |  | HWMId ≤1 | 1 < HWMId ≤ 3 | 3 < HWMId ≤ 6 | HWMId > 6 |
| GSOD | HWMId ≤1 | - | 501 | 143 | 40 |
|  | 1 < HWMId ≤ 3 | 894 | 172 | 53 | 17 |
|  | 3 < HWMId ≤ 6 | 321 | 101 | 30 | 15 |
|  | HWMId > 6 | 149 | 42 | 19 | 10 |
|  |  |  |  |  |  |
|  | Sensitivity | 0.69 | 0.21 | 0.12 | 0.12 |

Table 1 Confusion matrix of heat wave detection from observation (GSOD) and reanalysis (ERA-INTERIM) for maximum temperature for the period 1981-2015.

|  |  | **ERA-INTERIM** | | | |
|---|---|---|---|---|---|
|  |  | HWMId ≤1 | 1 < HWMId ≤ 3 | 3 < HWMId ≤ 6 | HWMId > 6 |
| **GSOD** | HWMId ≤1 | - | 344 | 61 | 10 |
|  | 1 < HWMId ≤ 3 | 920 | 117 | 31 | 4 |
|  | 3 < HWMId ≤ 6 | 357 | 58 | 12 | 5 |
|  | HWMId > 6 | 177 | 36 | 10 | 2 |
|  |  |  |  |  |  |
|  | Sensitivity | 0.70 | 0.21 | 0.11 | 0.10 |

Table 2 Confusion matrix of heat wave detection from observation (GSOD) and reanalysis (ERA-INTERIM) for minimum temperature for the period 1981-2015.